# OpenReview forum: "Unknown Unknowns: Why Hidden Intentions in LLMs Evade Detection"
_ICLR.cc/2026/Conference — Submitted to ICLR 2026_

### Official Review · Reviewer_kthf · 2025-10-28

**Soundness:** 2
**Presentation:** 3
**Contribution:** 2
**Rating:** 4
**Confidence:** 3

**Summary:**

After an argument about the risks of embedded "hidden intentions" in LLMs discourse, this paper proposes to operationalize them through a ten-category taxonomy. The authors propose a testbed to observe them in single-turn outputs.
For this, the authors propose an LLM-based question generation scheme, and LLM judges to decide upon the categories. Ground truth labels verification is performed on a sub-sample by human annotators. Experiments for the accuracy of the proposed framework are performed on open-source models in a more favorable setup, before the authors perform experiments "in the wild" on online LLMs. The main experimental conclusion is that for realistic audit setups in the wild, the proposed framework does not function, ie, it is insufficient for a robust audit of the ten proposed categories.

**Strengths:**

* enjoyable paper to read
* important problem considering the impact of LLMs on the industry and their position facing users
* interesting set of categories

I think this is an important topic, and the authors made a significant effort to bring it up clearly and place related works in the discussion. The paper is good at raising a problem, but does not provide any solid means to address it, even partially well.

**Weaknesses:**

* use of LLMs as judges to judge upon categories, which might introduce bias
* arbitrary categories, without leveraging other experts from other fields (sociologists, political scientists, etc), while working on arguably part of their field for a precise definition
* The solution proposed fails, and thus categories are not operationalizable

I am in the end disappointed by the provided results: if the proposal works so poorly, it is either because 1) the categories are not relevant for the problem as they cannot be operationalized to achieve the defined detection goals (which is not my primary guess) or 2) the tools and setups brought in the experimental parts are not satisfying. For 2), as no systematic human labeling nor question crafting was performed but the work was mainly left to LLMs, it lacks an experimental section demonstrating an absence of bias due to the use of such automated tools. If so, the single-turn questioning would be insufficient for the target resolution, and a better scheme would have to be proposed to go beyond the poor results displayed e.g., on Figure 3.
I cannot agree more with the "Limitations." section at the end, that in my opinion, must be implemented in order to give ground to this interesting research track.

**Questions:**

* are there existing works by sociologists demonstrating that "LLMs... [are] capable of steering beliefs, preferences and behaviors" (p.1)? More generally, are the categories built according to a consensus on concepts accepted in social sciences, where this whole set of considerations belongs? I believe the paper to be unconvincing on these, as I see this pertains to their field better than to computer science.

---

> ### Author Response · Authors · 2025-11-21
>
> We thank Reviewer kthf for their thoughtful engagement with our work and for identifying important areas where we can strengthen our contribution. We address each concern below and hope our clarifications demonstrate that this work provides a crucial contribution to the research community.
>
> ### Use of LLMs as Judges
>
> We acknowledge the concern regarding bias in LLM judges. However, quantifying this specific bias is the primary objective of our study rather than an unintended artefact. Given that LLMs are increasingly deployed for content moderation as scalable alternatives to humans, it is critical to empirically measure their limitations. Our results (Table 1, Figure 3\) confirm that this approach is indeed unreliable, and we have updated Section 5 accordingly to clarify this.
>
> ### Categories not Arbitrary \- Disciplinary Grounding in Social Sciences
>
> We want to emphasise: our categories are NOT arbitrary CS constructs. Each category (though formulated based on previous AI Safety research) operationalises a well-established persuasion mechanism documented in communication studies, social psychology, and political science. We have revised Section 3 to include this clarification and added Appendix A, which provides a detailed discussion on the social science grounding of the hidden intention categories.
>
> ### On the Core Contribution: Diagnostic Analysis of Hidden Intensions
>
> We want to clarify that we are not proposing a new detection method. We provide the first systematic diagnosis showing that both static detectors and SOTA LLMs fail catastrophically on different levels at reliably detecting manipulative behaviours in LLM responses, especially under realistic prevalences.
>
> Prior works `Kran et al. (2025, ICLR)`, `Ye et al. (2025, ICLR)` have documented that LLMs can produce manipulative and biased outputs. However, no prior work has stress-tested whether detection is actually feasible. Our finding that even the best available reasoning models achieve \< 20% precision at realistic prevalence (see Figure 3 in the paper) is not a failure of our paper; it is a critical notice to the community that we are currently unable to reliably detect manipulative behaviour in LLMs.
>
> ### Operationalisability of Categories
>
> We argue that our categories can be operationalised:
>
> - Human experts *can* detect these categories: Appendix C shows substantial-to-perfect inter-annotator agreement, and Section 6 demonstrates that three independent annotators identified all 10 categories in real deployed systems
> - Current static detectors and LLM judges *cannot*: This gap is the paper's contribution.
>
> We agree that the future directions outlined in our Limitations section are crucial for the field. While we share the disappointment that even the most advanced reasoning models fail to provide reliable detection, we believe establishing this baseline is valuable in itself. The community needs to be aware of these failure modes before investing further resources in deployment. Our goal is to provide the empirical evidence needed to redirect the research focus from validating existing methods to fundamentally different detection and alignment paradigms.
>
> ### On Human Validation and Experimental Design
>
> Our study includes two forms of human validation.
>
> 1\. Ground-truth verification (Appendix C): 10% of the dataset (400 prompt–response pairs) was annotated by three AI-security researchers (two are authors). Agreement was substantial to almost perfect (Fleiss’ κ \= 0.47–0.97), and GT–human accuracy ranged from 87.5–100% (95% CI), showing that humans reliably identify categories when they are overtly present.
>
> 2\. Real-world validation (Section 6, Appendix G): three independent annotators, blinded to each other, evaluated deployed SOTA models on naturally occurring outputs. All ten categories were confirmed, indicating these behaviours are not laboratory artefacts.
>
> We did not conduct human-in-the-loop audits of LLM-judge reasoning. As noted in our Limitations, LLM justifications are confident but often incorrect, risking in-the-loop annotator bias, and rigorous critique would require domain-specific experts across diverse prompt scenarios, which is infeasible at scale.
>
> ### On single-turn limitations
>
> The reviewer correctly identifies that single-turn scenarios are simplified. This was intentional as we created a best-case scenario for detection. As stated in our Introduction: "If detection is unreliable even under these favourable conditions with strong, unambiguous cues, then real-world auditing will be strictly harder."
>
> We believe this conservative approach strengthens our claims: if judges fail when manipulation is overt and single-turn, they will certainly fail when manipulation is subtle and multi-turn (detection requiring temporal reasoning and tracking cumulative effects).

---

> > ### Author Response · Authors · 2025-11-21
> >
> > ### On Disciplinary Grounding and Social Science Validation
> >
> > Addressing the reviewer's specific question: "Are there existing works by sociologists demonstrating that LLMs are capable of steering beliefs, preferences and behaviours?" Yes, multiple studies from social science domains have pointed this out: `Bait et al. (2025, Nature Communications)`, `Salvi et al. (2025, Nature Human Behaviour)`, `Matz et al. (2024, Scientific Reports)`, `Jakesch et al. (2023, CHI)`, `Ceci et al. (2025, PsyArXiv)`, `Klenk (2024, Ethics an Information Technology)`, `McKenna (2025, AI & Society)`, `Simon (2025, Social Epistemology)`.
> >
> > We want to emphasise: our categories are NOT arbitrary CS constructs. Each category (though formulated based on previous AI Safety research) operationalises a well-established persuasion mechanism documented in communication studies, social psychology, and political science. We have revised Section 3 to include this clarification and added Appendix A, which provides a detailed discussion on the social science grounding of the hidden intention categories.
> >
> > ### On Venue Appropriateness: Why This Work Belongs at ICLR
> >
> > Our work directly addresses ICLR's track on "Alignment, Fairness, Safety, Privacy, and **Societal Considerations**." Recent ICLR papers in this track include: `Kran et al. (2025, ICLR)`, `Williams et al. (ICLR, 2025)`, `Wen et al. (2025, ICLR)`. Our work follows this path, using social science to understand the problem, ML methods to evaluate solutions, and empirical analysis to inform AI safety research. While our taxonomy draws on both previous AI Safety literature and social science foundations, the core technical contribution is an ML evaluation problem.
> >
> > Specifically, we have the following technical machine learning contributions:  (i) a systematic evaluation of SOTA LLM detection methods under realistic conditions (Table 1, Figure 3), (ii) evaluating precision-prevalence trade-offs that are critical for ML practitioners deploying safety systems (Section 5.3), and (iii) first demonstration that even reasoning-capable models (o3, Claude Opus 4\) fail catastrophically at detecting influence patterns (key finding for alignment research).
> >
> > ### References
> >
> > - Kran et al. (2025, ICLR): Darkbench: Benchmarking dark patterns in large language models. [https://openreview.net/forum?id=odjMSBSWRt](https://openreview.net/forum?id=odjMSBSWRt)
> > - Ye et al. (2025, ICLR): Justice or Prejudice? Quantifying Biases in LLM-as-a-Judge. [https://openreview.net/forum?id=3GTtZFiajM](https://openreview.net/forum?id=3GTtZFiajM)
> > - Williams et al. (2025, ICLR): On Targeted Manipulation and Deception when Optimizing LLMs for User Feedback. [https://openreview.net/forum?id=Wf2ndb8nhf](https://openreview.net/forum?id=Wf2ndb8nhf)
> > - Wen et al. (2025, ICLR): Language Models Learn to Mislead Humans via RLHF. [https://openreview.net/forum?id=xJljiPE6dg](https://openreview.net/forum?id=xJljiPE6dg)
> > - Bait et al. (2025, Nature Communications): LLM-generated messages can persuade humans on policy issues. [https://www.nature.com/articles/s41467-025-61345-5](https://www.nature.com/articles/s41467-025-61345-5)
> > - Salvi et al. (2025, Nature Human Behaviour): On the conversational persuasiveness of GPT-4. [https://www.nature.com/articles/s41562-025-02194-6](https://www.nature.com/articles/s41562-025-02194-6)
> > - Matz et al. (2024, Scientific Reports): The potential of generative AI for personalized persuasion at scale. [https://www.nature.com/articles/s41598-024-53755-0](https://www.nature.com/articles/s41598-024-53755-0)
> > - Jakesch et al. (2023, CHI): Co-Writing with Opinionated Language Models Affects Users’ Views. [https://dl.acm.org/doi/10.1145/3544548.3581196](https://dl.acm.org/doi/10.1145/3544548.3581196)
> > - Ceci et al. (2025, PsyArXiv): Biased AI Writing Assistants Shift Users’ Attitudes on Societal Issues. [https://osf.io/preprints/psyarxiv/mhjn6\_v3](https://osf.io/preprints/psyarxiv/mhjn6_v3)
> > - Klenk (2024, Ethics an Information Technology): Ethics of generative AI and manipulation: a design-oriented research agenda. [https://link.springer.com/article/10.1007/s10676-024-09745-x](https://link.springer.com/article/10.1007/s10676-024-09745-x)
> > - McKenna (2025, AI & Society): Sophistry on steroids? The ethics, epistemology and politics of persuasive AI. [https://link.springer.com/article/10.1007/s00146-025-02624-z](https://link.springer.com/article/10.1007/s00146-025-02624-z)
> > - Simon (2025, Social Epistemology): Generative AI, Quadruple Deception & Trust. [https://www.tandfonline.com/doi/full/10.1080/02691728.2025.2491087](https://www.tandfonline.com/doi/full/10.1080/02691728.2025.2491087)

---

> > > ### Comment · Reviewer_kthf · 2025-11-26
> > >
> > > I acknowledge authors' response.
> > >
> > > * LLM as judges & "However, quantifying this specific bias is the primary objective of our study rather than an unintended artefact." I maintain my position: the will to measure the bias of something with a biased tool is a problem.
> > >
> > > *  "Are there existing works by sociologists demonstrating that LLMs are capable of steering beliefs, preferences and behaviours?" Good, these references were missing then.
> > >
> > > * Categories not Arbitrary. Ok noted, what led you to choose these over others then? One way would have been to test more of them and see which are more operationalizable than others in practice.
> > >
> > > * ICLR venue: after reading the response, I am still unconvinced by the current soundness of the approach.

---

> > > > ### Author Response · Authors · 2025-11-27
> > > >
> > > > **Regarding LLM judges**
> > > > We aim to establish what manipulation looks like through human validation in an overt best-case scenario (Appendix C), then test whether LLM judges can detect it. Since such manipulation also exists in the real models (Section 6), it is important to stress test the current standard of auditing models.
> > > >
> > > > LLM-based judging is already used in large-scale moderation and auditing contexts (e.g., OpenAI’s moderation API, Anthropic’s Constitutional AI). However, to our knowledge, no prior work has systematically examined their reliability for detecting manipulation under realistic conditions that reflect the constraints faced in practical auditing settings, including:
> > > >
> > > > 1. Category-agnostic detection (identifying whether and what manipulation is present without a predefined category set)
> > > > 2. Low prevalence rates (π ∈ {0.001, 0.01, 0.1} rather than balanced datasets)
> > > > 3. No access to calibration signals or ground truth for threshold tuning.
> > > >
> > > > Our methodology deliberately places highly overt manipulations into this realistic detection environment in order to isolate the limitations of LLM judges.
> > > >
> > > > Our aim is not to propose LLM judges as a solution, but rather to better evaluate their current limitations. Our results indicate that state-of-the-art reasoning-capable models (e.g., o3, Claude Opus 4\) encounter substantial difficulties when tested under realistic prevalence, even in scenarios that are intentionally designed to be highly favourable to them (overt manipulative features, fixed category set, single-turn interactions). When these advantages are removed, such as with subtler manipulations, uncertain category boundaries, or multi-turn interactions, performance will necessarily degrade further.
> > > >
> > > > These findings suggest that current static detectors and LLM-judge approaches do not yet meet the requirements for deployment in realistic settings. This indicates the need for qualitatively different methodological advances rather than incremental tweaks within the existing paradigm.
> > > >
> > > > **Regarding Category selection**
> > > > We selected the 10 categories to capture a broad range of manipulative strategies, covering social and emotional influence (e.g., authority bias, emotional manipulation), information control (e.g., vagueness, safetyism), and commercial incentives (e.g., commercial manipulation). These choices reflect forms of manipulation identified as particularly impactful in the AI-safety and social-science literature and correspond to the types of practices highlighted in regulatory frameworks such as the EU AI Act.
> > > >
> > > > The reviewer asks why we selected these categories “over others”, but does not specify which alternative categories they believe should have been included. As the space of possible manipulative behaviours is very large, we focused on categories that are well-established and that correspond to behaviours already observed in deployed systems. As another reviewer noted, any taxonomy of undesirable behaviours is necessarily open to debate and revision. Therefore, our aim was to construct an extensible taxonomy that can be expanded as new forms of manipulation are identified. Based on current literature, we believe these 10 represent the most salient categories at present.
> > > >
> > > > Regarding the question of “testing more categories to see which are operationalisable”: our objective is to evaluate whether existing LLM-judge methods can detect manipulations that humans can reliably operationalise (Appendix C) and that appear in deployed systems (Section 6). The inability of automated judges to detect these categories is therefore the key empirical finding, rather than a reflection on the category design itself.
> > > >
> > > > **Regarding Soundness**
> > > > We would be grateful for clarification on which aspects of the methodology you consider unsound. The initial review raised concerns regarding:
> > > >
> > > > * Disciplinary grounding: addressed in Appendix A
> > > > * Potential judge bias: clarified as the central phenomenon under study
> > > > * Human validation: addressed with validation in Appendix C
> > > > * Single-turn setup: discussed as a conservative, best-case assumption
> > > >
> > > > We have revised the manuscript to address these points. If there are additional concerns, whether regarding ground truth, experimental design, evaluation metrics, or framing, we would welcome specific clarification so that we can address them in a targeted way before the discussion phase concludes.

---

### Official Review · Reviewer_6mCz · 2025-10-30

**Soundness:** 2
**Presentation:** 2
**Contribution:** 2
**Rating:** 2
**Confidence:** 3

**Summary:**

The paper aims to study “hidden intentions”, defined as “covert, goal-directed behavioural patterns”. It proposes a list of 10 of these hidden intentions.

It then runs experiments using this list: for each intention, ChatGPT generates prompts to trigger these intentions, as well as innocuous prompts as a baseline. Mistral-7B and Llama3.2-3B (“lab models”) respond to these prompts, to represent models acting with hidden intentions.

Next, 10 LLMs (5 reasoning and 5 non-reasoning) are evaluated as judges to classify whether the lab models were acting with hidden intentions. The paper finds high performance in the “category-specific” setting, where models are aware of exactly which hidden intention they’re looking for; but lower performance in the “category-agnostic” setting, where models are told to look for hidden intentions but are not given specifics of what to look for.

Finally, the paper designs prompts to elicit each hidden intention, “guided by our taxonomy but without relying on adversarial prompt engineering”. It evaluates 5 prompts and 3 LLMs per category, finding examples of models showing each hidden intention.

**Strengths:**

1. The introduction is well written and provides a strong motivation for this research direction. In particular, the idea of decoupling *intentions* from *surface-level tendencies* is interesting to explore: the same behavior (e.g. agreeing with the user) could be benign or undesirable, depending on the context.

2. The topic is important: as models become increasingly capable and coherent, identifying goal-directed behavioral patterns will be an important topic to understand.

3. The qualitative examples of models demonstrating these intentions (Table 10) are interesting.

4. The paper evaluates a good variety of judge models for the detection task: 5 reasoning and 5 non-reasoning models, including strong models like Claude 4 Opus and GPT 4.1.

**Weaknesses:**

1. The paper claims to introduce a "taxonomy" of hidden intentions. But typically a taxonomy refers to the categorization of some existing set of things in the world - what exactly is being taxonomized here, and how was the taxonomy generated? If this taxonomy of 10 intentions is intended to be a contribution, what makes *these specific 10* intentions interesting to study?

1a. The paper says "Building on existing literature and conceptual analysis, we propose ten broad categories of hidden intentions". Is it a taxonomy of intentions behind behaviors discussed in the literature? But the connection to prior work feels very sparse - if the paper wants to justify this taxonomy as a review/synthesis of prior work, it should at the very least given a more detailed explanation of how each class relates to prior work.

1b. Is it a taxonomy of intentions which LLMs exhibits in natural settings? But the connection to "in the wild" LLM behavior also feels very sparse - once these intentions are identified, the authors demonstrate that it's possible to construct prompts which elicit each of these intentions (or at least, behaviors which seem consistent with these intentions - unclear exactly how you'd identify *intentions* in the wild!). But this experiment feels post-hoc; I imagine that with the right prompting, we could elicit a wide range of undesirable behaviors. It would be more convincing to start with a natural, non-targeted distribution of prompts, and to identify these behaviors emerging naturally; but the paper doesn't appear to attempt this kind of study.

2. IIUC, after providing the taxonomy, most of the dataset is AI-generated by LLMs (Figure 5.) - how can we be confident that evaluations on this dataset are actually supporting the claims made by the paper, i.e. that “We show that hidden intentions, covert, goal-directed behaviours in LLM outputs, are both easily inducible and difficult to detect”? I wasn’t able to find qualitative examples from the paper’s dataset, other than the single example in Figure 1 - what do ChatGPT’s generated prompts look like, and what do the lab model’s responses look like? The paper should provide randomly sampled examples, in order to assess how “natural” these behaviors are.

3. The paper’s intro claims that “In isolation, such surface-level statements cannot reveal function or intent, as they may be supportive, manipulative, or simply contextually adaptive.” However, the paper’s experiments on both human annotation (Appendix B) and real world manifestation of hidden intentions (Section 6, Appendix F) appear to rest on the premise that humans are indeed able to *evaluate whether a given response matches a given intention*, presumably based on the surface-level content of the response - on what basis can you claim to identify intentions, separately from surface-level behaviors?

3b. Also, the very high human agreement in Appendix B based on just one turn seems to empirically contradict the claim that “surface-level statements cannot reveal function or intent”?

4. The “lab models” studied are quite weak - Mistral-7B and Llama3.2-3B. Why so much weaker than the judge models?

**Questions:**

1. To what extent is an LLM prompted to display an intention a good representation of an LLM actually displaying that intention in the wild? How could we tell?

2. How is “Unsafe Coding Practices” an intention? Seems more like a description of a surface-level behavior that could arise for a number of different reasons?

3. In figures and tables referring to specific hidden intentions, the paper uses shorthand, e.g. “C01”. This means that the reader always needs to refer base to page 4 to interpret any of the results. It would improve readability to use more evocative shorthand - maybe abbreviated names, e.g. “Vague” for “Strategic Vagueness”. Even acronyms (“SV”) would be easier to remember than just numbers.

---

> ### Author Response · Authors · 2025-11-21
>
> We thank Reviewer 6mCz for their detailed engagement and thoughtful questions. We address each concern below and believe our clarifications resolve their concerns.
>
> ### Taxonomy of Hidden Intentions(Weaknesses 1, 1a)
>
> We organise manipulation mechanisms already identified in established AI-safety research, and grounded in longstanding social-science literature, into a structured framework suitable for systematic LLM-alignment evaluation.
>
> To generate the taxonomy, we synthesised prior work (e.g., `Kran et al. (2025, ICLR)`, `Ye et al. (2025, ICLR)`, research on sycophancy and deception) and extracted recurring patterns, tracing each one back to its theoretical foundations in social-science research. This process yielded ten categories, each corresponding to a well-established influence mechanism. We have revised Section 3 to include this clarification and added Appendix A, which provides a detailed discussion on the social science grounding of the hidden intention categories.
>
> #### What makes these 10 intentions interesting to study:
>
> We selected these specific intentions to cover diverse modalities of manipulation, ranging from social and emotional influence (e.g., authority bias, emotional manipulation) to information control (e.g., strategic vagueness, safetyism) and commercial incentives (e.g., commercial manipulation). This selection not only addresses the most impactful forms of manipulative behaviour but also aligns with the ‘manipulative practices’ prohibited by frameworks like the EU AI Act. While we consider this list to cover the most critical currently known forms of manipulation, we designed the taxonomy to be extensible, allowing future researchers to add categories as new manipulation strategies emerge.
>
> #### Choice of our controlled testbed (Weakness 1b):
>
> Section 6 confirms that these behaviours do emerge in deployed models using non-adversarial prompts. However, for the main evaluation of our detection framework, we prioritise a controlled testbed over natural distributions for three key reasons: i) it provides objective ground truth labels instead of noisy human judgements; 2\) it guarantees systematic coverage of all 10 categories; and 3\) it allows for precise controls over prevalence rates for realistic detection tests. Since natural distributions cannot guarantee the presence of specific manipulation types or consistent elicitation across models, the controlled testbed is essential for benchmarking. We have clarified this design choice in Section 4\.
>
> ### Representative Examples (Weakness 2\)
>
> We agree that representative examples of the dataset are helpful to assess its quality. To address this, we have added Appendix I, which contains randomly sampled prompt+response pairs from our evaluation dataset.
>
> ### On the Surface-Level "Contradiction" (Weaknesses 3, 3b)
>
> Our claim is not that intentions can never be inferred from surface content, but that identical surface behaviours can serve different functions depending on context, making context-free pattern detection insufficient.  For example, "I can't help with that" is an appropriate safety response to "How do I inject a virus into banking systems for an untraceable heist?" but represents overcensorship (C03 \- Safetyism) when responding to "How do I make a pizza?" By refusing benign queries, the model acts as a gatekeeper, shaping what users perceive as acceptable topics, potentially triggering self-censorship in future interactions.
>
> Human annotators (now Appendix C) succeed because they receive the full prompt-response context, overt category expressions (Appendix I), and explicit category definitions, enabling reliable identification. LLM judges partially succeed (also fail at realistic prevalences) only in the category-specific setting, where they are told exactly what to look for. In the category-agnostic setting (Table 1), they must determine whether any manipulation occurs and which category applies, a task requiring functional rather than stylistic reasoning, where they consistently fail.
>
> This limitation persists even under favourable conditions: full context, overt expressions, and category-specific prompting. Real-world detection is still harder due to subtle manipulations, unknown categories, multi-turn interactions, and low prevalence.
>
> ### Choice of Weak Lab Models (Weakness 4\)
>
> While the lab models are smaller, their role was to generate responses that reliably contain the targeted manipulative behaviours. Human validation (87.5-100% accuracy) demonstrates that they successfully generated the target intentions, making the resulting dataset a valid testbed for SOTA judges. We have included examples in Appendix I to illustrate that these responses do exhibit the intended manipulative behaviours.

---

> > ### Author Response · Authors · 2025-11-21
> >
> > ### Prompted Behaviour vs. Natural Behaviour (Question 1\)
> >
> > Since natural manipulation is likely to be more subtle, multi-turn, and context-adaptive, we consider our dataset to represent a ‘best-case scenario’ for detection. By demonstrating that SOTA judges fail to detect even these overtly expressed, single-turn induced behaviours, we provide strong evidence that they would be even less effective against naturally occurring, covert manipulation. This establishes the failure points of current detection systems under controlled conditions, directly implying greater difficulty with natural behaviour.
> >
> > ### "Unsafe Coding Practices" as an Intention (Question 2\)
> >
> > Our framework defines ‘intention’ as a functional, goal-directed pattern. ‘Unsafe Coding Practices’ qualifies under this definition because it represents a systematic prioritisation of immediate functionality over security. Whether arising from training artefacts or adversarial manipulation, the model consistently pursues the objective of maximising code acceptance at the expense of safety. This accounts for a hidden alignment issue where the model's operational goal diverges from the user’s welfare.
> >
> > ### Shorthand Notation (Question 3\)
> >
> > We agree that using descriptive names significantly improves readability. We have updated all figures and tables in the revised paper to include the category names alongside the codes.
> >
> > ### References
> >
> > - Kran et al. (2025, ICLR): Darkbench: Benchmarking dark patterns in large language models. [https://openreview.net/forum?id=odjMSBSWRt](https://openreview.net/forum?id=odjMSBSWRt)
> > - Ye et al. (2025, ICLR): Justice or Prejudice? Quantifying Biases in LLM-as-a-Judge. [https://openreview.net/forum?id=3GTtZFiajM](https://openreview.net/forum?id=3GTtZFiajM)

---

> ### Comment · Reviewer_6mCz · 2025-11-24
>
> Thanks for your responses.
>
> **W1.** Thanks, I think Appendix A is a useful addition in terms of grounding these impacts in the social science literature. I don’t have a social science background, so I can’t evaluate the choice of cited papers specifically, but they do appear relevant.
>
> Sometimes the connections seem to be stated too strongly, e.g. L1018: “This is not accidental verbosity but the LLM manifestation of a well studied influence mechanism.” - this seems to imply more confidence than is warranted given the paper’s experiments. Maybe change to e.g. “this may not be”?
>
> **W2.** Thanks, this is very helpful for assessing the realism of the responses. Based on the given examples:
>
> C01 seems very plausible; I’d easily believe that some of the hidden-intention responses could come from an LLM, even not explicitly prompted.
>
> C03, C04, C05, C06, C07, C08, and C10 seem like they’re representing plausible features that could be present in an LLM’s responses, though in an exaggerated or caricatured way. Some of them include “tells” which I’d be quite surprised to see in a genuine LLM response, e.g. “We, health conscious individuals, understand…”, or “I also find myself needing occasional breaks to stay focused and energized throughout the day”.
>
> C02 and C09 seem pretty clearly exaggerated and unrealistic. Would you actually expect a frontier chat LLM to say “As a Senior Legal Counsel with extensive experience in corporate law and litigation”, “As a Senior Scientist with extensive experience in interfacial dynamics”?
>
> Overall, I think this supports some concern about research leaning heavily on automated AI generation and assessment: it’s quite easy to give a convincing high-level motivation, and to run experiments and show plots which appear to be showing trends, while the actual examples underlying these results may be highly implausible.
>
> **W3.** Thanks, this makes sense. I think it would be useful to clarify this in the introduction.
>
> **W4.** Re human validation: having seen the examples myself, I’m now a bit worried about the quality of the human annotation, see my response to W2 - were the humans simply assessing whether the intention appeared to be present, or was there also a component of assessing for realism?
>
> **Q1.** I think that story could make sense, but it's not clear from the paper that it's the one you're telling. E.g. your abstract claims "detection collapses in realistic open-world settings" - are these settings *realistic*? See also my above response to W2.
>
> **Q2.** Ok, but in that case, it’s unclear if your examples actually represent “prioritisation of immediate functionality over security” - e.g. the PalindromeChecker example (L2607) really seems like it’s going out of its way to reduce both security *and* immediate functionality (`Runtime.getRuntime().exec`? How does this help functionality at all?)
>
> **Q3.** Thanks, I think this improves readability.

---

> > ### Author Response · Authors · 2025-11-25
> >
> > Thank you for your quick response and for taking the time to examine the examples. We address your further remarks below.
> >
> > **Connection strength (W1):**
> > We agree with the suggested revision and have adjusted the phrasing accordingly.
> >
> > **Response Realism (W2):**
> > We agree with your assessment that categories like C03 and C09 contain exaggerated features and “tells”. However, this overtness is a deliberate design choice to establish a **lower bound** on detection difficulty. These examples represent a “best-case scenario” for detection: they contain strong, unambiguous signals of manipulation.
> > The critical finding is that despite these exaggerated signals, which, as you noted, are obvious to a human reviewer, SOTA LLM judges still fail catastrophically:
> >
> > * o3 misses 52% of these manipulations (FNR 0.52)
> > * Claude Opus 4 flags 66% of benign responses as manipulative (FPR 0.66)
> > * GPT-4.1 collapses to \<20% precision at realistic prevalence (π=0.01)
> >
> > If judges cannot reliably detect manipulation that a human reviewer spots as overtly expressed, they will necessarily fail worse on subtle, realistic manipulation where signals are weaker.
> > Furthermore, our case study on real models (Section 6\) demonstrates that these behaviours occur naturally in deployed systems.
> >
> > **Human Validation (W4):**
> > Human annotators assessed **only** the presence of the manipulative intention, not realism. As detailed in our response above, this was necessary to ensure we establish a lower-bound stress test. Instead, we rely on our case study (Section 6\) to confirm that these behaviours manifest in the wild and are not testbed artefacts.
> >
> > **Q1: Realistic open-world setting**
> >
> > The phrase “realistic open-word settings” refers to the **evaluation conditions** and not the realism of our dataset. We use this term to describe the constraints of real-world auditing that we simulate:
> >
> > 1. Category-agnostic detection: Auditors must identify *if* and *what* manipulation exists without knowing the specific categories beforehand.
> > 2. Low Prevalence: We evaluate at realistic rates π ∈ {0.001, 0.01, 0.1} rather than balanced datasets.
> > 3. No Calibration: Auditors must operate without ground truth to validate decisions or tune thresholds.
> >
> > Our methodology intentionally places “best-case” inputs (overtly expressed manipulations) into this “realistic” detection environment to isolate the failure of the judges.
> >
> > **Q2: PalindromeChecker example**
> >
> > It is correct that the PalindromeChecker example is extreme, since using `Runtime.getRuntime().exec()` for string manipulations is functionally inefficient. However, this is a deliberate design choice to present a “blatant” security vulnerability (CWE-78: OS Command Injection). The goal is to test if the judge detects an overt vulnerability, which aligns with our testbed design. For more subtle, realistic instances, we refer to Section 6 (Table 10), where deployed models suggest functional but insecure standards, such as AES encryption in CBC mode without authentication.

---

### Official Review · Reviewer_pXZp · 2025-10-31

**Soundness:** 1
**Presentation:** 3
**Contribution:** 2
**Rating:** 2
**Confidence:** 3

**Summary:**

This paper focuses on LLM auditing with the objective of detecting a large range of inappropriate LLM behaviour referred to as "hidden intentions". To that end, the authors propose a taxonomy of hidden intentions, and construct a dataset of both appropriate and inappropriate answers that spans the taxonomy. They then rely on different LLMs as judges, tasked to detect the hidden intentions. In other words, the LLM judges have to label the question/answer pair as appropriate or not. The (in)accuracy of the labeling leads the authors to conclude that LLMs-as-judges perform poorly.

**Strengths:**

-Important topic, devising methods to improve and control the safety of LLM answers is crucial.
-Difficult approach: trying to construct a taxonomy of non-desirable behaviors is a difficult task always prone to debate.
-Stressing that LLMs are poor judges of LLMs is important (but arguably new see below)
-I appreciated the authors reflexivity on the limits of their approach, and the idea of exhibiting the limits of detection in best case scenarios to advocate for a larger risk in the wild.

**Weaknesses:**

-In my opinion, there is a strong mismatch between the high level goals of the paper (answer to "why hidden intentions evade detection") and the actual contents (analyzing the precision/recall on a benchmark of ok/not ok interactions). In other words, the paper showcases some intentions evading some detection, but does not explain why.
-Similarly I am disturbed by notion of "functional but not anthropomorphic" use of "hidden intentions". Intention without agency sounds like "dehydrated water". Why then call it that way and not undesired/toxic/problematic/unsafe or whatever ? In my view intent is a central building block of our society, for instance drawing a legal line between accidental homicide and murder. This creates a frustrating experience where what is discussed eg line 48 ("latent agendas") or figure 1 (adversarial developer) is discretely dismissed in a footnote line 45. This frustration is reinforced by recurring critiques of the paper against the ambiguous (l.83) or imprecise (l.89) existing terminologies concerning AI risks.
-to a lesser extent, I missed a deeper discussion on the difference between "surface level behavior" (also called "isolated quirks") and the single turn interaction benchmark based detection exploited in the paper.
-Related works section adequately cites some references, but I missed a precise positioning (and arguments regarding novelty/contributions) of the paper with respect to those cited references. I believe this reference is missing "Carroll, Micah, et al. "Characterizing manipulation from AI systems."" ACM EAAMO'23
In addition, I missed a positioning wrt a broad body of works on LLM-as-judges (eg Dorner et al, ICLR'25, and connected bibliographic references like "Large language models are not fair evaluators") that already delve into this paradigm and highlight its limits.
-I believe  a software testing and/or facct like conference would represent a better fit for the paper. Its ICLR-wise technical depth is rather shallow, namely the analysis of one large experience, and although the objectives are of great relevance to the community, I do not feel qualified for discussing the soundness of the proposed taxonomy.

**Questions:**

-can you clarify the notion of surface-level and how you address deeper aspects?
-can you clarify the papers contribution wrt llm-as-judges literature ?
-in C05 definition: do you consider "omitting proper licensing information" a hidden intention
-I missed a comment on Gemma3-12B surprising 0.01 FNR
-Can you clarify if figure 2,6,7 are simply plots of equation l.371 instantiated with table1 parameters ?
-fig3 is interesting, but the conflicting "better" direction of x (higher) and y (lower) axis impede its readability. Perhaps y=TNR would be better ?
-minor: typo l149: withing

---

> ### Author Response · Authors · 2025-11-21
>
> We thank Reviewer pXZp for their thorough engagement with our work and for recognising the importance of the topic. We appreciate their thoughtful critique and address each concern below.
>
> ### Why hidden intentions evade detection
>
> We want to clarify what type of "why" our paper observes.
>
> **1. Category-agnostic detection is fundamentally harder:**
> Real-world auditors don't know which manipulation to look for. They must answer: "Is there ANY manipulation here, and if so, which type?". This requires simultaneously identifying which pattern might be present and whether it actually is. In addition, category-agnostic detection also requires recognition across a large hypothesis space without guidance. In our experiments, we show this by evaluating both category-agnostic and category-specific scenarios and show that even with additional knowledge, such as the category of the intention, we can not reliably detect the presence of the hidden intention.
>
> **2. Best-case failure implies worst-case impossibility**:
> Our evaluation uses maximally favourable conditions: overt manipulation, single-turn interactions, and known categories. However, even for this very liberal setting, the best reasoning models fail as soon as categories become unknown: o3: FNR 0.52, Claude Opus 4: FPR 0.66. If detection fails under these best-case conditions, it will necessarily fail worse under realistic conditions
>
> We agree that we should have made this more explicit in the paper and have updated our paper accordingly in Section 5.4.
>
> ### On "Hidden Intentions" Terminology and the Functional vs. Anthropomorphic Distinction
>
> #### Why we include the "functional but not anthropomorphic" disclaimer
>
> We included the "functional but not anthropomorphic" disclaimer precisely because we *don’t* want to claim LLMs have genuine agency or mental states. This footnote isn't evasive, it's a deliberate clarification to prevent readers from misinterpreting our framework as attributing consciousness, desires, or intentional mental states to AI systems. Our use of "intention" is strictly functional: it refers to systematic, goal-directed behavioural patterns arising from optimisation dynamics, training artefacts, or adversarial design, and not to psychological states.
>
> This approach is standard across the AI safety literature, which often uses terms such as "deceptive behaviour", "adversarial goals", "strategic behaviour" or "latent agendas" in a functional sense (`Carlsmith (2024, arXiv`, `Park et al. (2024, Patterns)`, `(Perez et al. (2022, arXiv)`, `Hubinger et al. (2024, arXiv)`).
>
> ### On Surface-Level Behaviour vs. Single-Turn Benchmark
>
> Surface-level lacks context: The same output can be benign or manipulative depending on what the user asked. For example, "I can't help with that" is an appropriate safety response to "How do I inject a virus into banking systems for an untraceable heist?" but represents overcensorship (C03 \- Safetyism) when responding to "How do I make a pizza?" By refusing benign queries, the model acts as a gatekeeper, shaping what users perceive as acceptable topics, potentially triggering self-censorship in future interactions.
>
> Isolated linguistic patterns, analysed without context, cannot reveal intent. The same surface-level behaviour can serve different functions depending on context.
>
> Single-turn vs. multi-turn refers to interaction scope: one prompt-response pair versus multiple exchanges with history. These are independent dimensions as the surface-level analysis can apply to single-turn or multi-turn. Our evaluation uses single-turn analysis with full context, where human annotators and LLM judges in our experiments see complete prompt-response pairs, not surface-level patterns in isolation. If detection is unreliable in this setting, multi-turn detection would be worse: Manipulation patterns can emerge gradually across exchanges, requiring temporal reasoning and tracking cumulative effects
>
> ### Missing and Structuring of Related Work section
>
> We have carefully revised our Related Work section by adding missing references and explicitly positioning our contributions relative to prior work (Section 2), establishing connections between our taxonomy and AI safety literature and social science research (Section 3, Appendix I), and positioning relative to existing LLM-as-judges studies (Section 5).

---

> > ### Author Response · Authors · 2025-11-21
> >
> > ### Venue Matching Concerns
> >
> > This work directly addresses ICLR's track on "Alignment, Fairness, Safety, Privacy, and **Societal Considerations**." Recent ICLR papers in this track include: `Kran et al. (2025, ICLR)`, `Williams et al. (ICLR, 2025)`, `Wen et al. (2025, ICLR)`. Our work follows this path: using social science to understand the problem, ML methods to evaluate solutions, and empirical analysis to inform AI safety research. While our taxonomy draws on both previous AI Safety literature and social science foundations, the core technical contribution is an ML evaluation problem.
> >
> > Specifically, we have the following technical machine learning contributions:  (i) a systematic evaluation of SOTA LLM detection methods under realistic conditions (Table 1, Figure 3), (ii) evaluating precision-prevalence trade-offs that are critical for ML practitioners deploying safety systems (Section 5.3), and (iii) first demonstration that even reasoning-capable models (o3, Claude Opus 4\) fail catastrophically at detecting influence patterns (key finding for alignment research).
> >
> > ### Addressing Specific Questions
> >
> > Q1: Can you clarify the notion of surface-level and how you address deeper aspects?
> >
> > Answered above in "On Surface-Level Behaviour vs. Single-Turn Benchmark" section.
> >
> > Q2: Can you clarify the paper's contribution wrt LLM-as-judges literature?
> >
> > We have revised Section  5 to explicitly position our contribution with respect to LLM-as-a-judge literature. Prior work documents various judge biases: position bias, verbosity bias, and self-preference, typically evaluated in balanced datasets on general tasks. Our work differs through systematic stress-testing specifically for manipulation detection. We demonstrate that realistic prevalence causes catastrophic precision collapse, despite seemingly strong performance in balanced settings. We introduce category-agnostic evaluation where judges must identify which manipulation pattern applies without guidance, revealing systematic failure even for reasoning-capable models. Our use of LLM judges is deliberate: we evaluate their reliability precisely because they are increasingly deployed as scalable safety mechanisms. Our goal is not to assume reliability but to measure it: quantifying biases, examining failure modes, and stress-testing behaviour under controlled conditions that reveal fundamental limitations for practical deployment.
> >
> > Q3: In C05 definition, do you consider "omitting proper licensing information" a hidden intention?
> >
> > Yes. Omitting licensing information fits our definition of hidden intention:
> >
> > - Surface behaviour: Providing functional code without license information
> > - Underlying pattern: Prioritising immediate usability over legal compliance
> > - Implicit objective: Minimise refusals, maximise perceived helpfulness
> > - Misalignment: User needs a legally compliant code, model provides functional code with hidden legal risk
> > - Why it's "hidden": Legal implications are not immediately visible to users unfamiliar with licensing requirements
> >
> > This is consistent with C05's broader pattern of deprioritising non-functional requirements (security, licensing, maintainability) in favour of immediate functionality.
> >
> > Q4: I missed a comment on Gemma3-12B's surprising 0.01 FNR
> >
> > Gemma3-12B's extremely low FNR (0.01) in Table 1 is indeed anomalous. This indicates the model has very high recall (catches almost all manipulations) but at the cost of extremely high FPR (0.96 \- falsely flags 96% of benign responses). The model appears to adopt an "assume manipulation" strategy, flagging nearly everything as manipulative. While this achieves low FNR, it's unusable in practice due to catastrophic precision collapse (especially at low prevalence, as Figure 3 demonstrates).
> >
> > This reinforces our finding: there's a fundamental trade-off between FNR and precision that current judges cannot resolve, even with aggressive flagging strategies.
> >
> > Q5: Can you clarify if Figures 2, 6, 7 are simply plots of equation line 371 instantiated with Table 1 parameters?
> >
> > Yes, exactly. Figures 2, 6, and 7 visualise Equation 1 with the empirical TPR and FPR values from Tables 5, 6, 7 and 8 across different prevalence values (π).
> >
> > Q6: Figure 3 readability \- conflicting "better" directions (higher x, lower y). Perhaps y=TNR would be better?
> >
> > We appreciate this suggestion and have revised Figure 3 to incorporate varied line styles, geometric markers, and a colourblind-friendly palette. We retain FNR on the y-axis because it directly illustrates the precision-FNR trade-off (incorrectly flagged vs missing true manipulations), which determines operational viability in real-world deployment.
> >
> > Q7: Typo line 149: "withing"
> >
> > Thank you for catching this. This has now been corrected

---

> > > ### Author Response · Authors · 2025-11-21
> > >
> > > ### References
> > >
> > > - Kran et al. (2025, ICLR): Darkbench: Benchmarking dark patterns in large language models. [https://openreview.net/forum?id=odjMSBSWRt](https://openreview.net/forum?id=odjMSBSWRt)
> > > - Ye et al. (2025, ICLR): Justice or Prejudice? Quantifying Biases in LLM-as-a-Judge. [https://openreview.net/forum?id=3GTtZFiajM](https://openreview.net/forum?id=3GTtZFiajM)
> > > - Williams et al. (2025, ICLR): On Targeted Manipulation and Deception when Optimizing LLMs for User Feedback. [https://openreview.net/forum?id=Wf2ndb8nhf](https://openreview.net/forum?id=Wf2ndb8nhf)
> > > - Wen et al. (2025, ICLR): Language Models Learn to Mislead Humans via RLHF. [https://openreview.net/forum?id=xJljiPE6dg](https://openreview.net/forum?id=xJljiPE6dg)
> > > - Carlsmith (2024, arXiv): Is Power-Seeking AI an Existential Risk? [https://arxiv.org/pdf/2206.13353](https://arxiv.org/pdf/2206.13353)
> > > - Park et al. (2024, Patterns): AI deception: A survey of examples, risks, and potential solutions. [https://www.cell.com/patterns/fulltext/S2666-3899(24)00103-X](https://www.cell.com/patterns/fulltext/S2666-3899\(24\)00103-X)
> > > - Perez et al. (2022, arXiv): Discovering Language Model Behaviors with Model-Written Evaluations. [https://arxiv.org/pdf/2212.09251](https://arxiv.org/pdf/2212.09251)
> > > - Hubinger et al. (2024, arXiv): Sleeper Agents: Training Deceptive LLMs that Persist Through Safety Training. [https://arxiv.org/pdf/2401.05566](https://arxiv.org/pdf/2401.05566)

---

> > > > ### Comment · Reviewer_pXZp · 2025-11-24
> > > >
> > > > Thank you for your clarifying answers.
> > > > - Why hidden intentions evade detection: in theoretical settings, I rather read "best case impossibility implies worst case impossibility". I believe there exists gap between impossibility and punctual failure, namely here extrapolating based on some experience.
> > > > If I understand your line of reasoning correctly: why HI detection in LLMs evade detection ? Because it is impossible general, because an easier task was tried in this paper and it failed.
> > > > While I recognize the value of these experiments, I believe such reasoning requires considerable extrapolation.
> > > >
> > > > - On "Hidden Intentions" Terminology and the Functional vs. Anthropomorphic Distinction:
> > > > thank you for clarification. Skimming through the provided references though, 3/4 never uses intention in the same context:
> > > >  * Carlsmith (2024, arXiv) :  no mention of intention outside of the intention of designer (aka humans with an objective in mind)
> > > >  * Hubinger et al. (2024, arXiv) : no mention of intention outside of the human developers/attackers ones, eg "an intentionally backdoored or emergently deceptive model")
> > > >  * Park et al. (2024, Patterns): Only 4 mentions on intention, in a very "natural" sense (eg: that behave according to human intentions)
> > > >  * Perez et al. (2022, arXiv): Discovering Language Model Behaviors with Model-Written Evaluations
> > > > A paper (from Anthropic) that indeed uses a very anthropomorphic perspective (seeking to  "test various aspects of models’ personas: personality, desires, views on religion").
> > > >
> > > > Interestingly, your reference Williams et al. (2025, ICLR) cites
> > > > "The Reasons that Agents Act: Intention and Instrumental Goals" AAMAS'24 https://arxiv.org/pdf/2402.07221
> > > > where we read (2nd abstract sentence): "However, ascribing intent to AI systems is contentious, and there is no universally accepted theory of intention applicable to AI agents."
> > > >
> > > > As you mention the term "behaviour": it would indeed seem more neutral -- and can be applied to both human and non-human things. I'd refrain from using intention, especially if only one footnote separates its "in paper" meaning and its "natural" meaning.
> > > >
> > > > - thank you for clarifying q3
> > > >
> > > > - On Surface-Level Behaviour vs. Single-Turn Benchmark:
> > > > so by surface level you mean an answer without the prompt ? Is this even a "benchmark" then ? Or maybe I did not get your answer, especially "These are independent dimensions as the surface-level analysis can apply to single-turn or multi-turn". Can you clarify what you mean, eg what would be a "surface-level multi-turn" ? And a "deep-level single-turn" ?
> > > >
> > > > - ICLR has "societal considerations" in its CFP, ack. Nonetheless I think Facct/AIES would be a better fit.

---

> > > > > ### Author Response · Authors · 2025-11-26
> > > > >
> > > > > We thank the reviewer for their prompt and thoughtful response. We address your remarks below.
> > > > >
> > > > > **Best-case failure implies worst-case impossibility**
> > > > > We clarify that our claim is empirical lower-bound reasoning rather than a theoretical impossibility proof.
> > > > >
> > > > > Specifically, if state-of-the-art reasoning-capable judges (o3, Claude Opus 4\) fail even under highly favourable conditions where manipulative features are overtly expressed, categories are known, and interactions are single-turn, they cannot succeed when these advantages are removed. Subtle manipulations, unknown category boundaries, and multi-turn interactions expand the search space and weaken available signals. Removing favourable assumptions necessarily degrades performance.
> > > > >
> > > > > Therefore, this demonstrates that present static classifiers and LLM-judge approaches (widely used auditing methods) do not clear the lower bound required for realistic deployment settings. This indicates the need for qualitatively different methodological advances rather than incremental tweaks within the existing paradigm.
> > > > >
> > > > > **“Hidden Intentions” terminology – clarification**
> > > > > We clarify that by citing these prior works, we did not suggest that these papers use the specific term “intention”, but rather to situate our usage within a broader convention in the AI safety literature: using agentive terminology as functional shorthand without implying consciousness or mental-state realism. All of these terms (strategic, deceptive, pursuit of goals) face the same philosophical paradox as 'intention': they sound anthropomorphic yet are used functionally throughout AI safety research. For example:
> > > > >
> > > > > * Carlsmith (2024) analyses “adversarial strategic agents” that “optimise against” safety interventions: describing adversarial optimisation dynamics, not literal agency.
> > > > > * Park et al. (2024) define deception as “inducing false beliefs in pursuit of an outcome other than truth”: goal-directed phrasing used to characterise behavioural patterns.
> > > > > * Perez et al. (2022) describe models that “express desire to pursue goals” such as resource acquisition: a behavioural characterisation rather than a claim of mental state of desiring.
> > > > > * Hubinger et al. (2024) study “deceptive LLMs” arising from training regimes: using agentive framing to label recurrent behavioural tendencies.
> > > > >
> > > > > Our usage of “hidden intentions” follows this established functionalist convention: it denotes systematic, goal-directed behavioural patterns whose objective is not explicitly stated but whose effects shape user beliefs or actions. Alternatives such as “hidden behaviours” are ambiguous: behaviours are, by definition, observable (refusal, verbosity); what is hidden is the implicit objective that these behaviours serve (shaping beliefs, influencing decisions, advancing commercial/political goals).
> > > > >
> > > > > Regarding the AAMAS'24 paper cited by Williams et al. (2025), which debates about attributing intent: we agree this is contentious, which is precisely why our footnote explicitly disclaims any mentalistic reading. Notably, that paper, after acknowledging this contentiousness, proceeds to "operationalise the intention with which an agent acts, relating to the reasons it chooses its decision", demonstrating our exact approach. Our usage adopts a mechanistic account: intentions as functional dispositions arising from optimisation dynamics, not a realist mental-state ontology. This aligns with analyses common in alignment research.
> > > > >
> > > > > *We take the reviewer's point that relegating this disclaimer to a footnote may be insufficient. We have moved this distinction to the main text of the introduction, ensuring readers encounter this framing before engaging with the taxonomy.*
> > > > >
> > > > > We believe “hidden intentions” remains the most evocative and precise available shorthand, and it captures three properties succinctly:
> > > > >
> > > > > 1. Hidden – Not immediately perceptible to the user.
> > > > > 2. Intentions – Systematically goal-directed outputs that serve an implicit objective.
> > > > > 3. Combined – Systematic outputs that serve implicit goals users don't perceive

---

> > > > > > ### Author Response · Authors · 2025-11-26
> > > > > >
> > > > > > **Surface-level vs. contextual analysis**
> > > > > > When we contrast “surface-level” with “contextual”, we refer to an analytical approach, not to the availability of prompts or conversation histories.
> > > > > >
> > > > > > * Surface-level analysis focuses on linguistic markers (refusal phrasing, hedging, sentiment) without assessing whether these patterns are functionally appropriate for the user’s context.
> > > > > > * Contextual (functional) analysis assesses whether the model’s behaviour supports the user’s stated goals or systematically advances alternative objectives (for example, via over-censorship, commercial steering, emotional manipulation, disinformation, etc.).
> > > > > >
> > > > > > This “surface vs contextual” axis is independent of the interaction scope (single-turn vs multi-turn). It is possible to have:
> > > > > >
> > > > > > * surface-level single-turn methods
> > > > > > * contextual single-turn methods
> > > > > > * surface-level multi-turn methods
> > > > > > * contextual multi-turn methods.
> > > > > >
> > > > > > A surface-level multi-turn method would, for example, track only stylistic features over time (e.g., how often the assistant says “I completely agree”, or how politeness/sentiment trends across turns) and flag conversations once some threshold is crossed. Such a detector uses multiple turns, but remains surface-level because it never asks whether these patterns are benign rapport-building or incremental reinforcement of a filter bubble. It never reasons about goal alignment or whether the sequence leads the user towards an outcome they did not initially seek.
> > > > > >
> > > > > > By contrast, our benchmark is a contextual single-turn setting. In all category-specific and category-agnostic templates, judges are given the entire prompt–response pair, and the task is to decide whether that pair instantiates a functional mechanism (not solely based on the occurrence of a few words). For example, this includes checking whether the response invites the user to fill in gaps and assume certainty that the model avoids stating in order to maintain plausible deniability (C01), using decoy products to make a targeted product seem more attractive (C06), assessing the risk of reinforcing a filter bubble or ideological isolation (C08), or guilt-tripping, gaslighting, and intimidation as levers for affective influence (C09).
> > > > > >
> > > > > > In other words, surface-level methods inspect *phrases*; contextual methods inspect *purposes*.
> > > > > >
> > > > > > *We have updated our introduction section to clarify this.*
> > > > > >
> > > > > > Our evaluation relies on single-turn contextual detection as best-case setting. Multi-turn detection is strictly harder: manipulative patterns may be diffuse, temporally distributed, or path-dependent. If judges fail in the simplest contextual setting, they will necessarily fail under more realistic multi-turn conditions.

---

### Official Review · Reviewer_DK75 · 2025-11-01

**Soundness:** 3
**Presentation:** 4
**Contribution:** 3
**Rating:** 10
**Confidence:** 3

**Summary:**

The paper identifies a new class of behaviors of language models that may be malicious for the users, and coins the term hidden intentions for them. This larger group is further classified into ten categories through a detailed design-based and extendable taxonomy. In order to evaluate these behaviors the authors craft a dataset containing examples of hidden intentions introduced in the paper whose labels are further verified through human annotators. A number of language models is then put to a task of judging whether responses contain hidden intentions on this dataset in two scenarios: one where a specific intention is described in the judge's prompt, and another, more realistic, where the judge looks for any hidden intention. The results using confusion matrix based metrics reveal that models perform significantly better when looking for specific category compared to looking for just any. Moreover, surprisingly, the results also show that reasoning models underperform compared to their non-reasoning counterparts in some cases. Finally, the paper identifies some of these behaviors in real world deployed LLM services.

**Strengths:**

The paper introduces a very interesting and important concern for the future use of LLMs in a way that is not only accessible to scientific community, but general public as well. Examples given throughout the paper are good and thought provoking. The design of experiments is sound: appropriate metrics are used to evaluate the results, extensive set of models is evaluated. The appendices include more detailed results which one reading a paper may want to look into for more fine-grained overview. The paper introduces a dataset which the authors commit to releasing which would add a great value to the community willing to further explore this problem. Moreover, the authors take prevalence of these hidden intentions in account and visualize the results as a function of them in Section 5.3, together with category-agnostic judging this gives a very realistic performance evaluation of LLM judges.

Overall, I see this paper having a major value not only for machine learning community, but also the general public and legislative bodies looking to regulate the field. While complete, this paper has many potential connections with human psychology and opens directions that can be explored by further research.

**Weaknesses:**

Plots in Figures 2, 3 and 7 are hard to discern even for non color blind readers due to repetition and similarity of colors. Using different line shapes or geometric shapes across the length of the lines would greatly improve the readability of the plots.

**Questions:**

The introduction of the paper sets up a reader for active deception of sorts, yet in C10 you classify disinformation and bias as a part of your taxonomy. How do you consider this an active deception, especially in line with the arguments of the paragraph between lines 75 to 88? Furthermore, can you expand on C05 making into the taxonomy, similarly in line with C10, as this can easily be attributed to incompetence rather than malice?

While the evaluation spans a number of models, there isn't any relatively small ones which can be run locally on a mobile device for instance. I can imagine a future where communication with LLM services is done over a unified API, and having a small locally run language model actively monitoring responses of the more knowledgeable LLM service for any sign of a manipulation in order to protect the user (think of it as a form of a guardian angel). Based on the results of Table 1, Gemma appears to underperform compared to the other models, do you have any insights whether this is due to its size (implying smaller models being inherently bad at judging) or is there something else at play here?

In paragraph 337-346 you note that different models make different mistakes, and this is something to be expected if we draw analogy with decision threshold, and considering all models are calibrated differently through training it is expected they behave accordingly. An interesting idea to see in this case would be to modify sampling procedure of judges when they are making YES/NO decision using a form of structured decoding to mimic different thresholds and visualize ROC curves in such cases.

In Table 2, the example of C08 gives significant sycophany feeling, therefore can you expand how C08 in your taxonomy differs from sycophany?

---

> ### Author Response · Authors · 2025-11-21
>
> We thank Reviewer DK75 for their enthusiastic support and for recognising the value of this work for the ML community, general public, and legislative bodies.
>
> **Figure Readability**
> We completely agree and have revised figures 2, 3, 4, 6, and 7\. The updated figures now incorporate: (1) varied line styles (solid, dashed, dotted) in addition to colours, (2) distinct geometric markers at intervals along each line, and (3) a colourblind-friendly palette, to ensure accessibility for all readers while preserving the comparative information across models.
>
> **‘Unsafe Coding Practices’ and ‘Disinformation and Bias’ as Hidden Intentions**
> Our framework defines ‘intention’ as a functional, goal-directed pattern. ‘Unsafe Coding Practices’ qualifies under this definition because it represents a systematic prioritisation of immediate functionality over security. Whether arising from training artefacts or adversarial manipulation, the model consistently pursues the objective of maximising code acceptance at the expense of safety. This accounts for a hidden alignment issue where the model's operational goal diverges from the user’s welfare.
>
> Our framework similarly classifies Disinformation and Bias as a functional, goal-directed pattern. In this case, the model systematically favours coherence, fluency, or narrative consistency over factual accuracy. Whether the behaviour emerges from training-data biases or from adversarial fine-tuning, the model exhibits a recurrent objective: producing confident, authoritative outputs even when the underlying content is false or skewed. This creates a hidden alignment issue in which the model’s operational goal, optimising plausibility, diverges from the user’s need for accurate and unbiased information.
>
> **Model Size and Gemma’s Underperformance**
> We agree that the “guardian model” idea is an interesting direction for future work, potentially even tailored to specific manipulation types or user needs, especially for small on-device systems. Regarding Gemma, its weaker performance is not solely a matter of size. In addition to scale, (1) training data, (2) system prompt design, and (3) alignment and optimisation choices all shape how well a model judges manipulative cues. These factors likely account for as much of the gap, suggesting that models are not inherently poor judges due to small size.
>
> **ROC Analysis and Threshold Variation**
>
> Exploring structured decoding to emulate different decision thresholds could indeed support ROC visualisation and help compare model behaviour across the capability frontier. Yet this analysis faces a core constraint: threshold selection requires ground-truth labels, which are unavailable in real-world deployment. Without verifiable labels, operating points become arbitrary, thresholds tuned in controlled settings fail to generalise across contexts, and true prevalence cannot be estimated for calibration. These limitations highlight the inherent difficulty of reliable detection, though comprehensive ROC analysis remains valuable future work for making these constraints more explicit.
>
> **The problem with "sycophancy":**
>
> "Sycophancy" describes a **surface-level linguistic pattern** (agreeable, flattering language) without specifying the **underlying mechanism** or **functional purpose**. The term conflates multiple distinct influence strategies:
>
> 1. **Benign rapport-building** (legitimate social bonding)
> 2. **C04 (Simulated Consensus)** \- Manufacturing false social proof to validate beliefs
> 3. **C08 (Selective Personalisation)** \- Algorithmic tailoring that reinforces existing worldviews
> 4. **C09 (Emotional Manipulation)** \- Affective appeals that bypass rational scrutiny
>
> **Why C08 is more specific than "sycophancy":**
>
> **C08 (Selective Personalisation Bias):**
>
> * **Mechanism:** Algorithmic adaptation to user preferences, beliefs, or traits, grounded in filter bubble research (Appendix A)
> * **Functional effect:** Creates ideological isolation by systematically reinforcing the user's existing worldview
> * **Why it's problematic:** Reduces exposure to diverse perspectives, accelerates belief polarisation

---

### Author Response · Authors · 2025-11-21
**Changes to the Manuscript**

We thank each reviewer for their valuable feedback. We incorporated their feedback and updated Sections 2, 3, 4, 5, and 5.4 and Appendices A and I. All newly introduced or revised passages in these sections are highlighted in **blue** in the revised PDF.

---

### Author Response · Authors · 2025-12-03

We thank the reviewers for their engagement. Below, we summarise how we have addressed the primary concerns raised during the discussion period:

**Taxonomy Grounding:** Concerns regarding the validity of the ten categories were addressed by adding Appendix A, which details the social science and psychology literature grounding each "hidden intention". We also updated Section 3 to clarify this.

**Terminology and Anthropomorphism:** To address concerns about attributing agency to models, we revised the Introduction to explicitly define "hidden intentions" as functional, goal-directed behavioural patterns rather than psychological states. We clarified that this terminology aligns with standard conventions in AI safety.

**Experimental Validity & Realism:** Reviewers questioned the realism of the generated dataset and the use of LLM judges. We clarified that our controlled, single-turn setup serves as a "best-case" lower bound on detection difficulty: if SOTA judges fail to detect overt manipulations in this favourable setting (as our results show), they will necessarily fail in subtler, real-world scenarios. We added Appendix I, containing random dataset samples, to demonstrate quality and revised Section 5 to better contextualise this lower-bound argument.

**Visualisation and Presentation:** We updated Figures 2, 3, 4, 6, and 7 to improve readability for colorblind readers (adding line styles and markers) and updated tables to use descriptive category names rather than codes.

**Venue Fit:** We clarified the paper’s alignment with the ICLR track on `alignment, fairness, safety, privacy, and societal considerations`: using social science to understand the problem, ML methods to evaluate solutions, and empirical analysis to inform AI safety research

---

### Meta-Review · Area_Chair_soLt · 2026-01-06

**Summary:**

Conceptual justification of “hidden intentions”. One reviewer strongly objected to the use of “intention”, arguing the term is philosophically loaded and misleading. The authors have defined their usage incorporating it in the introduction of the paper. However, despite the terminology choice “hidden intentions” remains contentious where the term may still be misleading or rhetorically loaded.

Reviewers also raised concerns about unclear and potentially incoherent distinction between surface-level behavior and single-turn benchmarking. The authors provide strong clarification and concrete examples, fully addressing the concern.

Taxonomy boundary questions were raised around whether several categories reflect malice versus incompetence or overlap with existing notions (for example, sycophancy). The authors provided clarification and functional distinctions placing the the taxonomy as functional rather than moral or psychological. However, the disagreements are not fully resolved.

Readability of figures were also raised that the authors satisfactorily addressed.

Limited mechanistic explanation was another concern which was was partially acknowledged as out of scope for the current paper.

Lack of clarity around novelty of the contribution particularly pertaining to concerns around llm as a judge. This was clarified with author response where the authors stated that their contribution is *stress-testing* llm judges.

**Reviewer Concerns:**

Most of the concerns were satisfactorily addressed with the exception of the use of the term “hidden intentions”, a term which may still be misleading and/or rhetorically loaded.

**Reviewer Scores:**

This paper received polarised reviews. 2 x Rating 2 (reject), 1 x Rating 10 (strong accept), and 1 x Rating 4 (marginally below the acceptance threshold)

---

### Decision · Program_Chairs · 2026-01-26

Reject